# A binding protein regulates myosin-7a dimerization and actin bundle assembly

Rong Liu [1], Neil Billington [1], Yi Yang [1,2], Charles Bond[1], Amy Hong[1], Verl Siththanandan [1], Yasuharu Takagi[1] & James R. Sellers [1✉]

Myosin-7a, despite being monomeric in isolation, plays roles in organizing actin-based cell protrusions such as filopodia, microvilli and stereocilia, as well as transporting cargoes within them. Here, we identify a binding protein for *Drosophila* myosin-7a termed M7BP, and describe how M7BP assembles myosin-7a into a motile complex that enables cargo translocation and actin cytoskeletal remodeling. M7BP binds to the autoinhibitory tail of myosin-7a, extending the molecule and activating its ATPase activity. Single-molecule reconstitution show that M7BP enables robust motility by complexing with myosin-7a as 2:2 translocation dimers in an actin-regulated manner. Meanwhile, M7BP tethers actin, enhancing complex's processivity and driving actin-filament alignment during processive runs. Finally, we show that myosin-7a-M7BP complex assembles actin bundles and filopodia-like protrusions while migrating along them in living cells. Together, these findings provide insights into the mechanisms by which myosin-7a functions in actin protrusions.

[1] Laboratory of Molecular Physiology, National Heart, Lung and Blood Institute, National Institutes of Health, Bethesda, MD 20892, USA. [2] Present address: Laboratory of Functional Proteomics, College of Veterinary Medicine, Hunan Agricultural University, 410128 Changsha, Hunan, China. ✉email: Sellersj@nhlbi. nih.gov

Class-7 myosins are one of the most ancient myosins in the animal kingdom. They participate in fundamental processes from adhesion in *Dictyostelium* to sensory perception in humans[1–6]. Mammals and *Drosophila* each have two myosin-7 isoforms, myosin-7a and myosin-7b, which display distinct distributions and functional properties[7–9]. Defects in myosin-7a lead to deafness and retinal degeneration in mammals[10,11]. In the cochlea, myosin-7a localizes to actin-rich stereocilia in hair cells and regulates morphogenesis and mechanotransduction[7,12–14]. In the retina, myosin-7a is enriched in the microvilli of retinal pigment epithelial (RPE) cells and is required for melanosome entry into apical processes[15]. Myosin-7a is also present in secretory epithelial cells[2,16], where it distributes and anchors secretory granules in the actin cortex[17]. *Drosophila* myosin-7a exhibits similar distributions and cellular functions to its mammalian ortholog. It localizes to microvilli in the ovarian follicular epithelium[18], as well as to the actin-bundles of sensory bristles and the auditory Johnston's organ[5,19]. Mutations of *Drosophila* myosin-7a cause embryonic/larval lethality, with a fraction of adult "escapers" displaying hearing loss, infertility, and bristle abnormalities[5,19,20]. Due to the high degree of conservation, *Drosophila* has used productively as a model organism for understanding the structure, function, and regulation of myosin-7a[5,19,21,22].

All myosin-7 isoforms consist of a catalytic motor domain, a neck region with 5 IQ motifs, and a regulatory tail. The tail harbors a SAH domain followed by two MyTH4-FERM domains separated by an SH3 domain[5,22,23]. Purified myosin-7a is monomeric, with the C-terminal FERM domain docking against the motor to form a compact, autoinhibited molecule[22,24]. Myosin-7a belongs to a specialized subgroup of myosins, which have one or two MyTH4-FERM domains in their tails[6,25]. None of the known MyTH4-FERM myosins are natively dimeric, but several have demonstrated processive motility along actin when artificially dimerized[21,26–30]. MyTH4-FERM myosins are best known for their cellular localization to actin-bundle structures such as filopodia, microvilli, and stereocilia, and they have been shown to play essential roles in the development and function of these protrusions[6,12,19,31–33]. Despite this, detailed insights into the mechanisms and how they cooperate with interacting proteins to accomplish such tasks are still lacking.

Here, we uncover a *Drosophila* myosin-7a binding protein M7BP, and demonstrate how M7BP assembles a dimeric myosin-7a complex to enable cargo transportation and actin-bundle assembly. Given the structural–functional conservations of MyTH4-FERM myosins, our results have implications for the mechanistic understanding of prototypic class-7 myosins and other related myosin classes.

## Results

**Identification of a myosin-7a binding protein M7BP**. To search for proteins interacting with myosin-7a, we used the C-terminal FERM domain as bait to screen against a universal *Drosophila* cDNA library in yeast-two-hybrid assays (Fig. 1A). Of 40 robust colonies which grew, 25% had sequence matching an uncharacterized gene CG43340 in Flybase. CG43340 (also known as CG30492) contains 21 exons, associated with 15 transcriptional variants, predicted to code for 11 unique proteins. The myosin-7a interacting sequence is encoded on the C-terminal end and is distinct from any previously described myosin-7a binding protein[14,34]. The 5-prime region of the gene encodes a ~120 amino acid segment predicted to bind to Rab-proteins by structure-based sequence analysis (Supplementary Fig. 1A, B)[35]. Although none of the gene products were annotated, genetic screens showed that CG43340 is vital for *Drosophila*

embryogenesis[36–38]. In oocytes and embryos, CG43340 mRNA was found spatially enriched by dynein-BicD/Egl transport machinery for localized translation and polarity establishment[38]. During neurogenesis, CG43340 was activated[36] and involved in founding photoreceptor specification and sensory organ development[37]. The protein isoform C produced by CG43340 was found enriched in fly heads in mass spectrometry analyses[39]. We therefore selected isoform C for further study and termed it M7BP (myosin-7a binding protein). Immunostaining showed that M7BP co-exists with myosin-7a in larva's eye imaginal disks (Fig. 1B) and in affinity pulldown assays, M7BP was detected in the adult fly heads and was pulled out by purified myosin-7a protein that was immobilized on FLAG-resins (Supplementary Fig. 1E). M7BP is comprised of the putative Rab-binding domain N-terminally, a central region and the identified myosin binding domain (MBD) C-terminally (Fig. 1A). Although homology searches in mammalian databases using the MBD sequence alone do not find a direct homolog, searches using the whole sequence show the greatest homology to rabphilin, melanophilin, and MyRIP, suggesting that M7BP belongs to a wider family of Rab-dependent myosin binding proteins. Based on the domain organization, we predict that it may serve as an adapter protein between myosin-7a and Rab-protein associated cargoes, similar to MyRIP in regulating myosin-7a in humans[34].

**M7BP binds to myosin-7a to relieve its state of autoinhibition**. To directly study the interactions, we produced and purified full-length myosin-7a and M7BP using the baculovirus/Sf9 system. Purified myosin-7a holoenzyme is a monomer, containing one heavy chain associated with light chains of *Drosophila* calmodulin and ELC/Mlc-c, but not *Drosophila* RLC/sqh (Fig. 1C)[24,40]. Densitometry analysis suggested that the stoichiometric ratio between strongly bound calmodulin and ELC/Mlc-c is ~4:1. In addition, we found a subset of less tightly bound calmodulin that dissociated from heavy chain upon ultracentrifugation (Fig. 1C). While its binding region and biological significance remain unclear[41], no difference was detected in subsequent actin gliding or single-molecule motility assays when 1 μM excess calmodulin was added. M7BP is composed of 938 amino acids with a molecular weight of 103 kDa. Actin cosedimentation assays were performed for initial binding characterization (Fig. 1C). Myosin-7a alone did not sediment under high centrifugal forces but pelleted with F-actin in the absence of ATP. M7BP mostly remained in the supernatant either when centrifuged alone or with actin. There was near complete pelleting of M7BP when myosin-7a and actin were both present, demonstrating a strong interaction with myosin-7a. Replacing the myosin-7a with *Drosophila* myosin-5 in the actomyosin complex resulted in a dramatic reduction of the pellet-bound fractions of M7BP, suggesting a class-specific binding to myosin-7a. Using Bio-layer interferometry (BLI) we quantitatively measured the binding strength between myosin-7a and M7BP in the absence of actin, and found strong interaction with $K_d$ ~ 6 nM at approximately physiological ionic strength (Supplementary Fig. 1C, D).

Electron microscopy (EM) revealed that M7BP adopts a globular conformation at physiological ionic strength, with a size and asymmetry consistent with a monomer (Fig. 1D). We confirmed previous findings that myosin-7a with ATP and at physiological ionic strength exists in an autoinhibited conformation (Fig. 1E)[22]. The tail region folds back, contacting the motor. The strongly preferred orientation of molecules bound to EM grids results in these features being apparent even in the global average of all aligned molecules (Fig. 1E). With M7BP present, the appearance becomes more heterogenous but the same preferred orientation of the motor-lever remains, revealing the

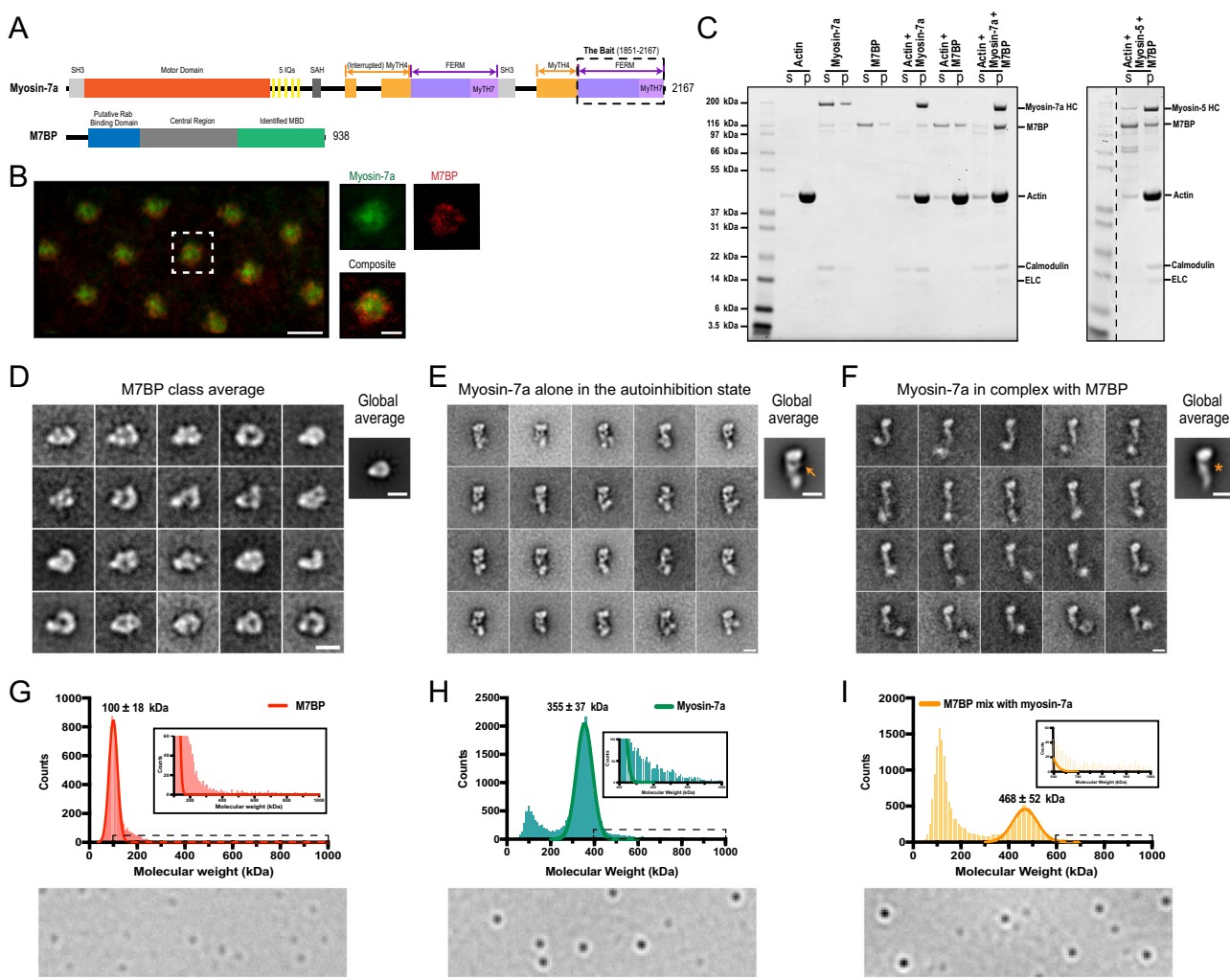

**Fig. 1 M7BP is a bona fide binding partner for myosin-7a that relieves its state of autoinhibition. A** Domain organizations of myosin-7a and the identified binding protein M7BP (Schematics are in scale with the length of their primary sequences). Numbers indicate the positions of amino acid residues. **B** Immunofluorescence staining of eye imaginal discs of third instar larvae with anti-myosin-7a (green) and anti-M7BP (red) antibodies. Scale bar = 5 μm in the left panel and 2 μm in the right panels of magnified images. **C** SDS-PAGE analysis of actin co-sedimentation assays. Fractions of supernatant (S) and pellet (P) isolated from mixtures of actin, myosin-7a, M7BP, or myosin-5 after ultracentrifugation were indicated. **D** Class averages from negative-stain EM images of M7BP in 150 mM salt buffer, showing an asymmetric globular structure. Scale bars = 10 nm. **E, F** Class averages from EM images of myosin-7a alone (**E**) and myosin-7a in complex with M7BP (**F**) in 150 mM salt buffer with ATP. Myosin-7a alone exhibits a compact, autoinhibited conformation in which the distal end of the folded tail contacts the motor domain (see orange arrow in the global average image). In the presence of equimolar M7BP, the complex exhibits a more extended conformation with concomitant loss of contact between the tail and the motor domain (orange asterisk), indicating a release of autoinhibition due to M7BP binding. Scale bars = 10 nm. **G–I** Interferometric single-molecule mass photometry assays. Histograms (bin width = 10 kDa) of single particle mass values (presented as molecular weight) for the indicated protein or protein complex. Lines are the Gaussian fit to the data yielding the molecular weights of samples. The small peak at ~100 kDa mass in **H** was attributed to multiple proteolytic products around this size that were also shown by SDS-gels in **C**. Insets: Zoom of the regions for larger species. Bottom: A singe frame of differential interferometric scattering images in each assay. N = 9063 M7BP molecules (**G**), N = 25,773 myosin-7a molecules (**H**), and N = 18,826 myosin-7a + M7BP complexes (**I**). Molecule landing events were collected over three independent experiments.

absence of the density contacting the motor (Fig. 1F). Due to the release of the flexible tail from its motor-attached position, the molecule is poorly resolved beyond the lever, compared to myosin-7a alone. Class averages are seen with additional density beyond the lever region, which is consistent with the size and shape of M7BP (Fig. 1F). In raw images, the tail is flexible and extended, with additional mass corresponding to M7BP attached at the distal end of the tail (Supplementary Fig. 1F, G). The predominant complex species under these conditions contains a single myosin molecule. Whilst higher order structures were occasionally apparent in EM images, these were rare and lacked the uniformity required for successful alignment.

Single-molecule mass photometry[42,43] enables measurement of myosin-7a-M7BP oligomeric species mass distributions. M7BP alone forms a single mass species of approximately 100 kDa, consistent with the EM images showing monomers (Fig. 1G). Myosin-7a alone produces a major species of 355 kDa (Fig. 1H), matching the mass sum of one heavy chain (250 kDa) plus six light chains (102 kDa). Addition of M7BP (two-fold stoichiometry) resulted in drastic reduction of free myosin-7a coupled to the appearance of a 468 kDa species, corresponding to the mass of a 1:1 complex of myosin-7a and M7BP (Fig. 1I). Despite large numbers of analyses (n > 18,000), we did not observe Gaussian-like distributions corresponding to higher-order oligomers (insets

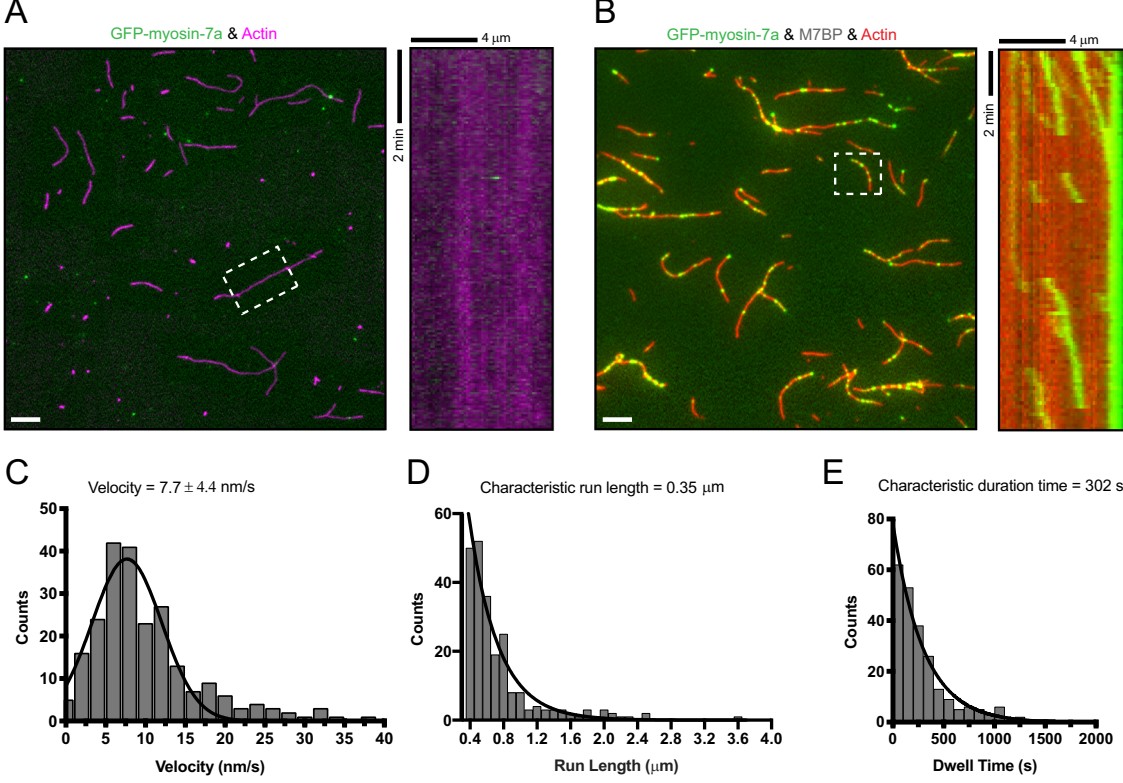

**Fig. 2 M7BP enables highly processive motility of myosin-7a along actin filaments. A** Left panel: A single time frame of GFP-myosin-7a alone on immobilized actin filaments (labeled with Alexa-Fluor 647 phalloidin, magenta). Right panel: Kymograph analysis of the box region in **A** showing that no processive movement was detected. Note the transient appearance of GFP-myosin-7a, which is indistinguishable from its binding to the background surface. Scale bar = 5 μm. **B** Left panel: A single frame taken from the time-lapse recordings of GFP-myosin-7a (green) with unlabeled M7BP. Actin is labeled with Rhodamine phalloidin (red). Right panel: Example kymographs (box region in **B**) show clear processive movements along actin filaments. Scale bar = 5 μm. **C–E** Frequency distribution histograms of the velocity (mean ± SD) (**C**), characteristic run length (**D**), and characteristic run duration (**E**), respectively for the GFP-myosin-7a in the presence of unlabeled M7BP. N = 231 processive tracks.

Fig. 1G-I). The near complete absence of free myosin-7a species is consistent with the high affinity of M7BP for myosin-7a.

M7BP moderately enhanced the steady-state ATP turnover at low concentrations of actin (Supplementary Fig. 1H), but had no effect on ATPase rates of skeletal-muscle myosin (Supplementary Fig. 1I). Together with EM data, the results show that M7BP activates ATPase activity in myosin-7a by binding to the C-terminal tail and disrupting formation of autoinhibitory head-tail associations.

**M7BP induces highly processive motility of myosin-7a along actin filaments**. To assess whether M7BP regulates myosin-7a's motility, we performed single-molecule motility assays using TIRF microscopy. In the presence of 150 mM KCl and 2 mM ATP, full-length myosin-7a alone transiently binds to the surface without measurable net motion (Fig. 2A). In striking contrast, we observed that myosin-7a binds to and moves along actin filaments with robust processivity when M7BP is present (Fig. 2B and Supplementary Movie 1). Detailed motility characterization shows that myosin-7a moves on actin slowly (~7.7 nm/s), and sustains long distance, long duration runs (0.35 μm and 302 s for run length and duration time, respectively, Fig. 2C–E). The results demonstrate that M7BP enables myosin-7a to move processively along actin filaments, strongly suggesting M7BP-induced oligomerization of myosin-7a during this process.

**Processive myosin-7a complexes are predominantly dimeric**. In early experiments, we observed that myosin-7a alone could move

for short distances upon addition of ATP if they were first clustered on actin in nucleotide-free conditions (Supplementary Fig. 2), indicating that myosin-7a may form homo-oligomeric complexes when brought into close proximity. Thus, we first asked whether motile complexes are myosin-7a self-assembled following M7BP-induced tail unfolding and whether M7BP remains associated with moving complexes. We engineered an mCherry-tagged M7BP and imaged it concurrently with GFP-myosin-7a. The two proteins were strongly colocalized in almost all motile puncta, and moved simultaneously along actin during the entire lifetime of processive runs (Fig. 3A, B and Supplementary Movie 2). By contrast, no binding or movement was detected for M7BP alone (Supplementary Fig. 3A), indicating that M7BP does not strongly interact with actin in the absence of myosin-7a under these conditions. These data suggest that M7BP is constitutively required in the complexes, likely engaged as a structural component along with myosin-7a tail domains for the establishment of an oligomeric interface.

To define more precisely the molecular composition, we examined the numbers of myosin-7a and M7BP per moving complex using quantitative single-molecule analyses. Initially, we determined the fluorescence intensity of single GFP or mCherry fluorophores by stepwise-photobleaching, and compared the values to unbleached complex intensity to extrapolate the molecule numbers[44]. Moving complexes predominantly contained two myosin-7a molecules, with a small subset of puncta having fluorescence intensities consistent with higher numbers of myosin-7a (Fig. 3C). For M7BP, the frequency distribution of fluorophore numbers was slightly left-shifted, with 1–2 molecules

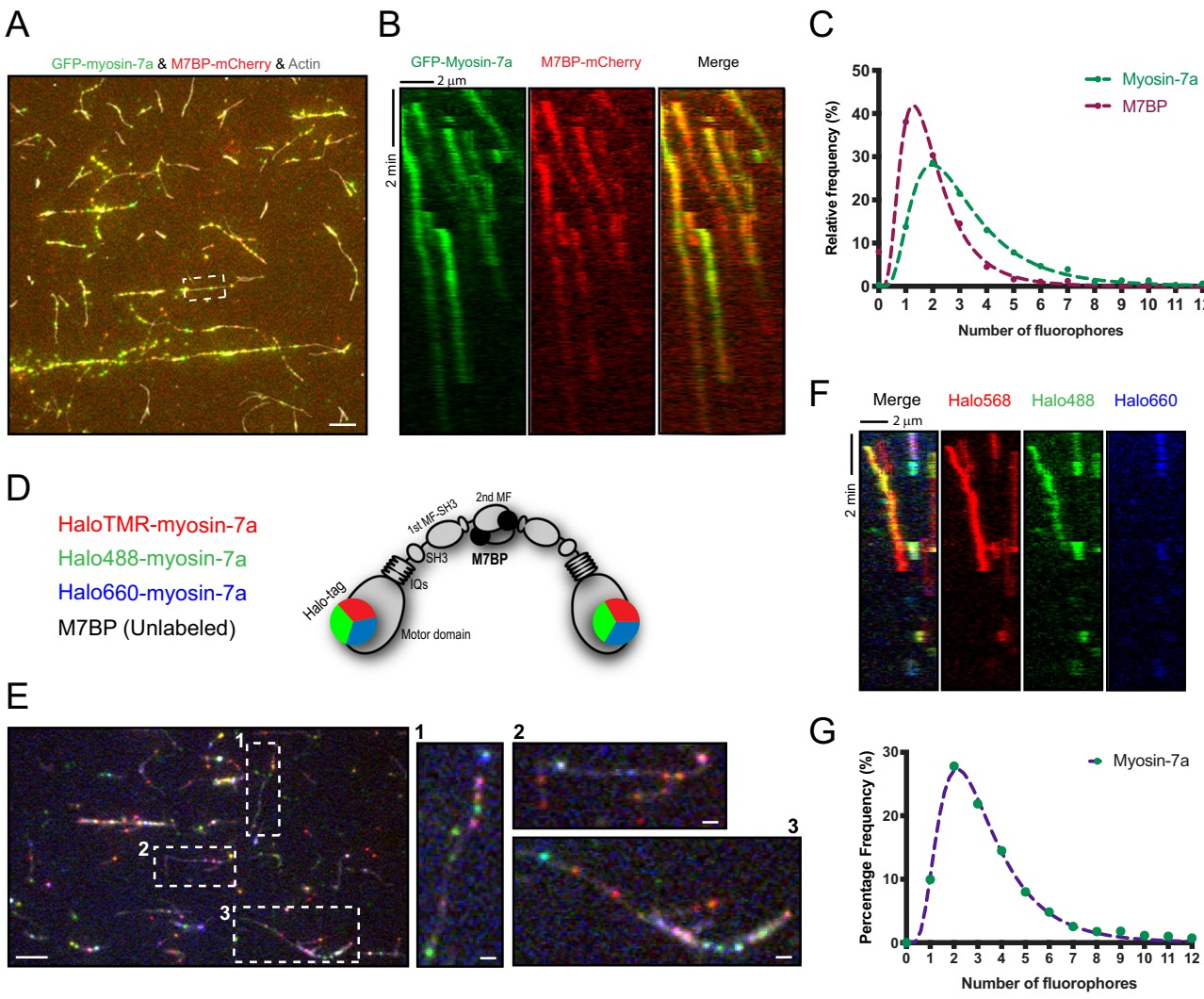

**Fig. 3 Processive myosin-7a complexes are predominantly dimeric. A** TIRF view of single-molecule motility assays with mCherry-tagged M7BP demonstrates that GFP-myosin-7a (green) and M7BP-mCherry (red) are colocalized in the majority of the puncta moving along actin filaments. Actin filaments were labeled with Alexa-Fluor 647 phalloidin (gray). Scale bar = 5 μm. **B** Kymograph analysis of box in **A** shows that the processive movements (diagonal lines) of GFP-myosin-7a and M7BP-mCherry are overlaid. **C** The histograms of fluorophore numbers (GFP for myosin-7a and mCherry for M7BP) determined by fluorescence intensity analysis. N = 661 quantified complexes. **D** Schematic of the three-color labeling experiments to test the hypothesis that the motile complex is composed of dimerized myosin-7a. Halo-tagged myosin-7a was labeled with TMR (red), Alexa-Fluor 488 (green), or Alexa-Fluor 660 (blue) and equal quantities of all three labeled myosins were simultaneously introduced into the flow chamber. **E** Movie frames show that myosin-7a labeled with three different colors moves processively along actin with low motor unit numbers (as most puncta exhibit the original or a mixture of two colors). Actin is labeled with Alexa-Fluor 405 phalloidin and displayed in gray). Unlabeled M7BP is present at the concentration equal to that of total myosin-7a. Images 1, 2, and 3 correspond to the regions in box 1, 2, and 3 at particular time points. Scale bar = 5 μm (left panel); Scale bar = 1 μm in images of 1, 2, and 3. **F** Kymograph analysis of the three-color single-molecule assays showing examples of different-color fluorophores moving together. **G** Frequency distribution histogram of the numbers of HaloTag fluorophores in each complex indicating that the motile complexes predominantly contain two myosin molecules. N = 1485 quantified complexes.

per complex being most common (Fig. 3C). It should be noted that mass photometry found a fraction of M7BP-mCherry (~20%) had masses consistent with proteolytic loss of mCherry tags (Supplementary Fig. 3B, C), therefore fluorophore counts for M7BP represent slight underestimates of the true molecule numbers. Considering inherent imprecision of this analysis due to factors such as optical noise, proximity of complexes, and inability to distinguish between final photobleaching events and complex detachments, we also adopted a strategy of labeling HaloTag-myosin-7a in three colors to more directly test for the presence of multiple myosins per complex (Fig. 3D). This approach not only allowed us to reliably distinguish the event of single-fluorophore photobleaching from the detachment of whole

complex by examining signals from other channels, but also provided direct evidence showing that the motile complexes predominantly contain low numbers of myosins (as puncta frequently observed in original or a mixture of two colors) rather than a large molecular assembly (as would result in bright white puncta) (Fig. 3E, F). The three-color single molecule analysis confirmed our observations from singly labeled myosin, together with the results from M7BP, showing that while higher subunit numbers are present, the most common structure adopted by the processive complexes are 2:2 myosin-7a-M7BP dimers (Fig. 3G).

**M7BP tethers actin filaments**. While no apparent interaction with actin was detected for M7BP alone, evidence from multiple

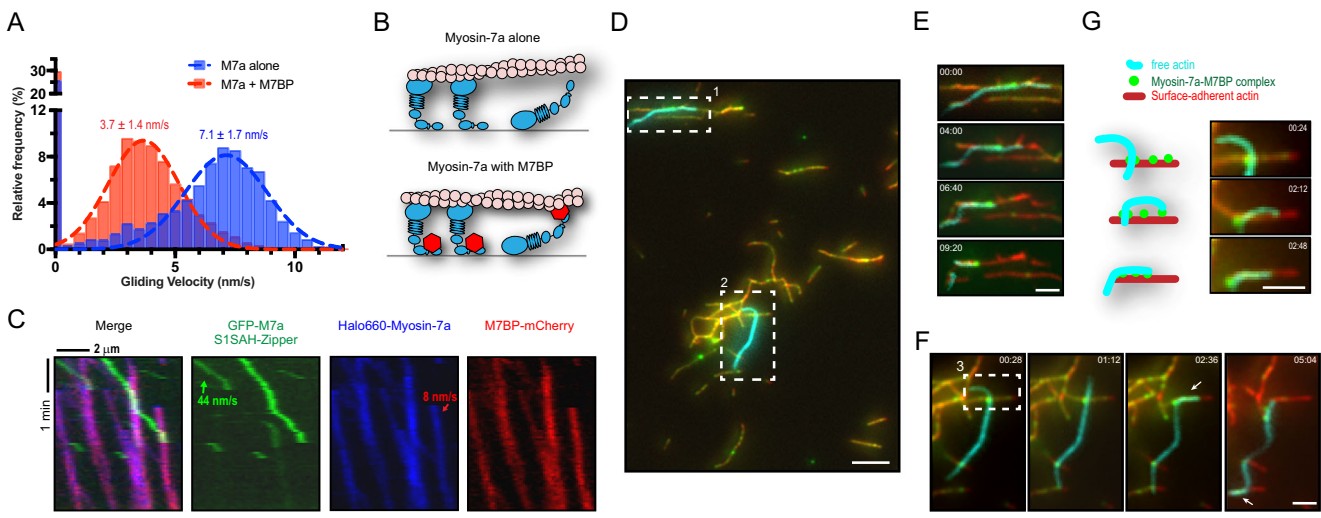

**Fig. 4 M7BP tethers actin filaments and enables myosin-7a-M7BP complex to drive actin-filament alignment in vitro. A** Speed distribution of actin movement by ensembles of surface-immobilized myosin-7a in the absence or presence of M7BP. $N = 6044$ actin filaments for myosin alone over four independent experiments. $N = 7467$ actin filaments for myosin with M7BP over four independent experiments. **B** Cartoon diagrams how infused M7BP tethers actin filaments and reduces the actin gliding speed. **C** Motilities of GFP-myosin-7a zippered dimer (green) and the myosin-7a-M7BP complex (shown in blue and red for myosin-7a and M7BP respectively) on the same actin filaments (unlabeled). Note that the zippered dimer of myosin-7a moves faster. **D–G** Myosin-7a-M7BP complexes tether actin filaments and actively drive them into alignment. Myosin-7a-M7BP complex is visualized by a GFP tag on myosin-7a (green). Immobilized actin filaments are labeled with Rhodamine-phalloidin (red) and free actin filaments labeled with Alexa-647 phalloidin (cyan). **D** A TIRF field view shows free actin filaments were bound to the surface by myosin-7a complexes. Scale bar = 5 μm. **E** Free actin filaments were transported by myosin complexes while they were moving along immobilized filaments (region of box 1 in **D**). **F** The two ends of a free actin filament were driven into alignment with surface-immobilized filaments against deformation strain (white arrow, region of box 2 in **D**). **G** Time-lapse images and cartoons showing that a free actin-filament end was caught by myosin-7a complexes, which caused the filament to bend and to align with the immobilized filament that the myosin was walking on (magnified time-lapse images of box 3 in **F**). Scale bars = 2 μm applied to **E–G**.

experiments suggested that M7BP is capable of binding to actin filaments in the presence of myosin-7a. Firstly, M7BP was found to inhibit actin-filament gliding over a myosin-7a-decorated surface in gliding motility assays. In 150 mM KCl, myosin-7a smoothly moved actin filaments with a speed of 7.1 nm/s. Infusion of M7BP into the flow chamber reduced the gliding velocity by approximately 50% and increased the proportion of tethered, static actin filaments (Fig. 4A). We speculate this is due to M7BP binding to the myosin tail and inhibiting the gliding via transiently interacting with actin (Fig. 4B)[45]. Indeed, actin-tethering by binding proteins is known to be a strategy used by myosins to enhance their transport efficiency and position cargoes onto cortical actin filaments[46,47]. Mechanistically, it was shown that actin-tethering slows overall moving speed but enhances processive runs by limiting diffusion from actin[48]. We therefore predicted that myosin-7a-M7BP complexes would move more slowly and processively than myosin-7a alone, if M7BP acts as a tether to the track. To test this directly, we constructed a GFP-tagged myosin-7a zippered dimer with the tail truncated to reveal its intrinsic motility in the dimeric state. Using single-molecule assays, we directly compared the movement between the zippered dimer and myosin-7a-M7BP complex under situations where they traveled simultaneously on the same actin filaments. As illustrated by kymographs (Fig. 4C), the myosin-7a-M7BP complexes move slowly and steadily, consistent with the results for the complexes characterized alone (Figs. 2–3). In contrast, the zippered dimers moved markedly faster, with shorter actin-attached lifetimes and processive runs (Supplementary Fig. 4). It is important to note that the traces for M7BP and the zippered dimer are entirely discrete, showing directly that M7BP does not bind the motor domain or neck regions but instead modifies myosin-7a's motility via bindings to the tails (Fig. 4C). Together, these results demonstrate that M7BP not only mediates the dimerization of myosin-7a, but also contributes towards processivity of the complex through its actin-binding activities.

**Myosin-7a-M7BP aligns actin filaments in vitro.** The findings that single myosin-7a-M7BP complexes can undergo long duration as well as multivalent interactions with actin led us to consider whether these complexes crosslink actin filaments laterally and lead to the formation of actin-filament bundles. To investigate this, myosin-7a-M7BP complexes were first added to surface-adhered actin filaments in the absence of nucleotide to allow decoration of filaments. We then infused free actin filaments and imaged dynamics of the actomyosin networks after addition of ATP. As shown in Fig. 4D, Supplementary Movie 3, free actin filaments were bound to surfaces with a tendency to overlap with immobilized filaments, confirming our predictions regarding the complex's actin-crosslinking capabilities. Relative movement was observed between filaments from separated layers, reflecting the dynamic nature of the motor-based crosslinking process (Fig. 4E, F). Indeed, unlike static crosslinkers that passively join neighboring filaments[49,50], myosin-7a-M7BP complexes were seen to simultaneously move and bind, producing active filament alignment as they collectively moved forward (Fig. 4F, G). Actin deformation and buckling were commonly seen if entanglements occurred, suggesting that the myosin-7a complex can exert alignment against mechanical constraints. Together, this in vitro reconstitution demonstrates that purified myosin-7a-M7BP complexes can actively drive alignment of actin filaments, providing a structural basis for recruitment of crosslinkers and bundle-tailored actin polymerases to stabilize and enhance the aligned filamentous network[50,51].

**Myosin-7a-M7BP complexes produce actin bundles and filopodial protrusions in living cells.** Finally, we expressed fluorophore-labeled myosin-7a and M7BP in *Drosophila* Schneider (S2) cells to determine the functional consequences of the complex in living cells. GFP-myosin-7a or M7BP-mCherry, when expressed individually, are diffuse throughout the cell with

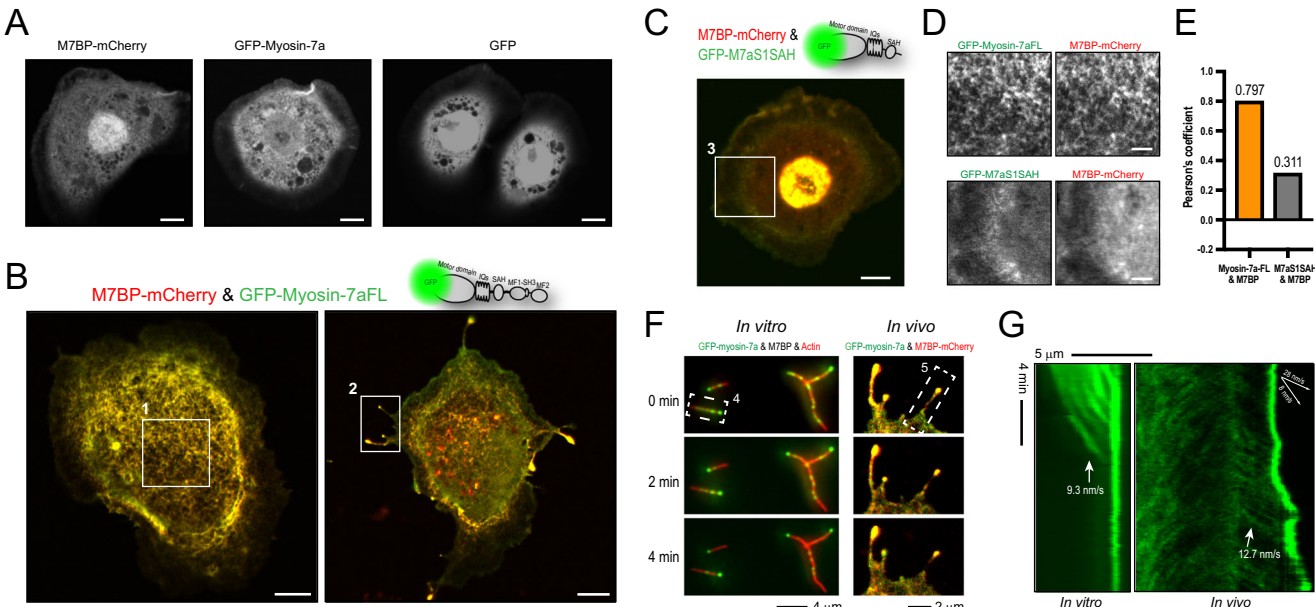

**Fig. 5 Expressing myosin-7a-M7BP complexes in cells produces substantial filamentous networks and filopodial protrusions. A** *S2* cells transfected with M7BP-mCherry or GFP-myosin-7a alone show no discernable difference in cell morphology compared to that of GFP transfection. Scale bar = 5 μm. (representative of five independent experiments) **B** Co-transfection of GFP-myosin-7a (green) with M7BP-mCherry (red) results in the induction of an extensive filamentous structure throughout the cell body and the extension of filopodial protrusions. Scale bar = 5 μm. Images are representative of 12 independent transfection experiments. **C** Co-transfection a monomeric tailless myosin-7a (GFP-M7aS1SAH, green) with M7BP-mCherry (red) does not induce the filopodia or the filamentous network phenotypes observed in **B**. Scale bar = 5 μm. (representative of four independent experiments) **D** Magnified images of box 1 in **B** (top panel) and box 3 in **C** (bottom panel) show that full-length myosin-7a and M7BP are colocalized on the filamentous structures, whereas M7aS1SAH and M7BP are diffusive and non-colocalized within the cell. Scale bars = 2 μm. **E** Colocalization analysis of regions from box 1 and box 3 yielded high degree of colocalization correlation between full-length myosin-7a and M7BP in contrast to the low colocalization score between M7aS1SAH and M7BP. N = 9802 pixels from ROI 6 × 12 μm. **F** Movie frames taken from in vitro single-molecule motility assays (left) and in vivo live cell imaging of a filopodium (right) show that myosin-7a-M7BP complexes stay for long duration at the end of actin filaments. **G** Kymographs of box 4 and 5 in **F** show that myosin-7a-M7BP complexes (green) travel with comparable speed in vivo along filopodia as they do in vitro along fixed actin filaments.

no alterations to overall cell morphology (Fig. 5A). Importantly, mRNAs for native myosin-7a and CG43340 in *S2* cells were at near-background or undetectable levels[52], hence the interactions of exogenously-expressed proteins with endogenous partners were considered minimal under these conditions. In striking contrast, we observed an extensive filamentous network formed throughout the cytoplasm, as well as extensions of filopodia-like protrusions when cells were co-transfected with both myosin-7a and M7BP (Fig. 5B). Myosin-7a and M7BP were colocalized along the filamentous structures and migrated together towards the filopodial tips (Supplementary Movie 4). Conversely, tailless myosin-7a and M7BP, which do not form a complex as shown in Fig. 4C, failed to colocalize or replicate these phenotypes but instead distributed throughout the cytoplasm (Fig. 5C–E). Simultaneous tracking of the filamentous networks with endoplasmic reticulum (ER), shows them to be distinct from the ER networks despite some morphological similarities (Supplementary Fig. 5A). Indeed, imaging with fluorescently-labeled actin demonstrates that these filamentous structures are actomyosin bundles assembled by myosin-7a-M7BP complexes (See details in Fig. 6). Kymograph analysis of the motion along filopodia revealed that the complex's moving velocity was in good agreement with the motility characterized in vitro (Fig. 5F, G).

To define more precisely which myosin domains are required for reorganizing the actin cytoskeleton we created myosins with deletions or point mutations. Cells co-expressing M7BP with a myosin-7a tail domain failed to induce cytoplasmic actin networks or filopodia, despite retaining strong colocalization of the two proteins (Supplementary Fig. 5B). Expression of myosin-

7a mutant RK/AA (resulting in constitutively active motor)[22] in the absence of M7BP similarly did not induce actin networks or filopodia, but reintroducing M7BP to RK/AA restored the phenotype (Supplementary Fig. 5C, D). Additionally, dimerized tailless myosin-7a induced numerous filopodia but not the interior filamentous structures (Supplementary Fig. 5E, F), showing that the tail domain along with associated M7BP is specifically required for the latter phenotype. Together, these results argue strongly that the complex's motor activities as well as the M7BP subunit are essential factors for actin network reorganization.

To further understand how myosin-7a-M7BP complex drives the cytoskeletal rearrangement, we performed TIRF structured-illumination microscopy to visualize spatiotemporal correlations between actin filaments and myosin-7a-M7BP complexes. The actin cytoskeleton in adhered *S2* cells was primarily organized as a well-defined lamella along the cell periphery[53]. Expression of myosin-7a-M7BP complex drastically reshaped the actin cytoskeleton with numerous actin filaments and bundles were seen throughout cell interior and formation of actin bundle-filled protrusions (Fig. 6A, B and Supplementary Movie 5). Importantly, the emergence of bundle-like structures was in temporal and spatial register with local accumulation of myosin-7a complexes (Fig. 6B, C), indicating a link between complex association and formation of actin-filament bundles. We were able to observe single myosin-7a complexes moving toward the tips within stable filopodia, and occasionally rearward motion of the complex during filopodial retraction and actin-bundle disassembly (Fig. 6D–F and Supplementary Movie 5). These

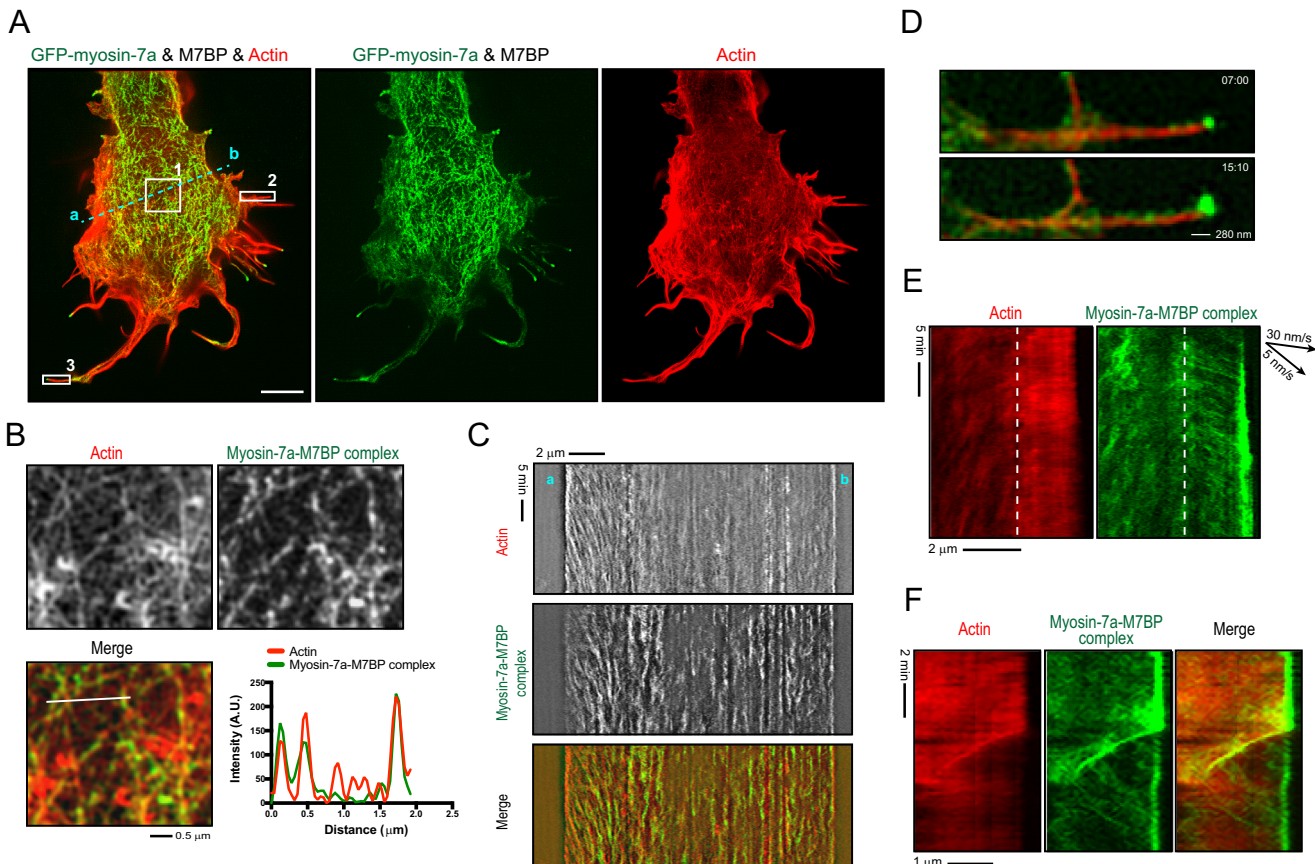

**Fig. 6 Myosin-7a-M7BP complexes drive actin filament bundling and filopodia extension. A** TIRF-SIM live-cell imaging reveals that expression of myosin-7a-M7BP complexes (visualized by a GFP tag to myosin-7a, green) produces substantial actin bundles in cell body and filopodial protrusions (actin was labeled with F-tractin-mCherry, red). Scale bars = 5 μm. Fluorescent Images are representative of three independent experiments. **B** Magnified insets of box 1 in **A** and line scans showing that actin networks is composed of densely packed actin filaments and bundles. Myosin-7a-M7BP complexes are primarily colocalized with actin bundles that thus results in a "filamentous appearance" of the complexes. **C** Kymograph analysis (cyan dash sample line in **A**) showing that myosin-7a complexes are associated with actin bundles during entire imaging period. Note a more vigorous actin retrograde flow was detected at one side of the cell edge (a), compared to the regions in cell body and in the opposite side (b) where actomyosin bundles are relatively static. Newly polymerized actin filaments on the cell edge (a) were associated with less myosin-7a complex. **D** Magnified images of box 2 in **A** showing single molecules of myosin-7a-M7BP complexes move along stable actin filament bundles, similar to the movement observed in vitro. Note the increase in tip intensity over time. **E** Kymograph analysis of box 2 in **A** reveals that while filopodial actin bundles remain stable, myosin-7a-M7BP complexes move forward to the tip of filopodia. The migration speed is comparable to the velocity measured in single-molecule assays or observed in live cells without actin labeling. The dash line indicates the cell edge. **F** Kymograph analysis of box 3 in **A** showing an example that filopodial bundles partially disassembled where a portion of the bundle retracted back towards the cell body. The associated myosin-7a complexes exhibit rearward movements along with the actin retrograde flow.

observations suggest that the complex's motion in filopodia is an interplay of plus-end motility with the actin turnover, which varies temporally during filopodial growth and retraction[54]. Additionally, inward flow of myosin-7a complexes was detected from the periphery to the center of the cells, likely as a result of actin retrograde flow behind cell edges (Fig. 6E). Collectively, the results demonstrate that the myosin-7a-M7BP complex assembles actin bundles and transports along actin-rich protrusions in cells.

## Discussion
Here, we uncover a previously undocumented binding partner for myosin-7a, and determine how the protein activates and co-assembles with myosin-7a to form a highly processive complex. We show that M7BP can tether actin when associated with myosin tails. By contrast, M7BP alone does not bind to or diffuse along actin filaments and its localization when expressed alone in S2 cells is diffuse. This suggests that M7BP may itself exist in an autoinhibited state where actin binding is prevented. We have

previously shown and confirm here that myosin-7a is auto-inhibited in the absence of M7BP[22]. Thus, the complexation of myosin-7a and M7BP may regulate the actin-binding activities of both proteins.

Further, the complex-actin interaction appears to regulate the assembly of the complex itself. EM and mass photometry show that isolated myosin-7a-M7BP complexes are almost exclusively monomeric with respect to each component. This contrasts to the single-molecule motility analysis, which reveals a prevalent 2:2 myosin-7a-M7BP dimeric structure for moving complexes, with M7BP linking the myosin-tail to actin filaments. The higher order oligomerization therefore seems to require the presence of actin. While it remains to be determined how the complex dimerization is coupled with the association of actin tracks, one tempting speculation is that the tripartite interactions among myosin-tail, M7BP, and actin produce conformational rearrangements within the monomeric subunits to enable formation of dimeric motor-adapter complexes. This provides a more efficient mechanism for dimerizing motors than proximity-mediated dimerization in the

absence of binding partner, which we showed to be an inefficient process. In addition to the motile dimeric complexes we also observe moving complexes with more than two myosins. The existence of these higher order complexes, which can be seen directly in motility experiments using three separate colors of myosin, suggests that the process of oligomerization is distinct from other myosins such as mammalian myosin-5a, where a single strong dimerization interface (i.e., the coiled-coil) results in a homogenous population of dimers. Future work to determine high resolution structures for the components of the complex will be required to determine the nature of and the regulation of these interactions.

Besides translocation, myosin-7a-M7BP complex displays the remarkable ability to reorganize actin networks in reconstituted systems as well as in living cells. The prominent actin-bundle structures induced by co-expressing myosin-7a and M7BP implies that the complexes may enhance overall actin-bundling activity and contribute to the core stiffness of the protrusions[55,56]. Moreover, fascin, the main actin-crosslinking protein in the protrusions, is known to be highly bundle-selective and studies both in vitro and in *Drosophila* bristles have demonstrated that actin filaments must be aligned before fascin can stabilize the structures[49,50,57]. Although we do not envision this myosin complex itself being able to form ordered paracrystalline arrays in the presence of ATP, such as those seen with static crosslinking proteins, the complex's ability to drive actin alignment with its plus-end motility may, in collaboration with actin nucleators, arrange filaments in parallel to allow efficient recruitment of fascin and other proteins[58]. This type of actin rearrangement would therefore be distinct from that induced by static cross-linkers in that it is an active, ATP-dependent process. The loosely bundled actin created by such a process would then be primed for stabilization into the highly ordered structures which form over time via the repeated binding of static crosslinkers. In addition, tailless myosin-7a dimers were seen to induce filopodia, suggesting that the processive motility of the motors per se has roles in filopodia initiation[59]. Interestingly, along with the cytoplasmic bundles, we only observed moderate filopodia induction by myosin-7a-M7BP complexes compared to that by the tailless dimers. In view of the high stability of the interior actomyosin network, one possibility is that the substantial bundle assemblies may suppress actin turnover[60], and thus limit the actin polymerization peripherally and filopodial protrusions which depend on the concentration of free actin monomers[61,62]. The results also imply that myosin-7a is more suited to actin protrusions with low treadmilling rates but high rigidity, such as in stereocilia or bristles[63,64]. It is also important to note that besides protrusive bundles, myosin-7a localizes to actin-filament bundles within the cells, e.g., the nurse cell strut in *Drosophila*[18] and the connecting cilium of photoreceptor cells in mammals[65]. Further investigation will be required to understand how myosin-7a participates in different actin-bundle structures within the native cellular environment. The discovery of M7BP and the mechanisms described in this study shall prove to be useful for these inquiries, and for understanding the functions and regulations of MyTH4-FERM myosins in general.

## Methods

**Protein production, purification, and labeling**. cDNAs encoding for myosin-7a and M7BP were inserted into a modified pFastBac1 vector, which contains a FLAG-tag sequence to the C-terminus. For the preparation of fluorescence labeled molecules, GFP or HaloTag moiety were cloned to the N-terminus of myosin-7a and the mCherry-tag to the C-terminus of M7BP (Infusion, Clontech) (Supplementary table). Transposition and the generation of recombinant baculovirus were performed following manufacturer's protocols (Thermo Fisher Scientific). For producing myosin holoenzyme, Sf9 insect cells were co-infected with recombinant baculovirus encoding for myosin-7a heavy chains, *Drosophila* calmodulin and

*Drosophila* cytoplasmic myosin light chain (Mlc-c). The proteins were purified via Flag affinity chromatography (Sigma) as described[22] with minor modifications. Briefly, cell pellets were homogenized in extraction buffer containing 150 mM NaCl, 10 mM MgCl$_2$, 1 mM EGTA, 2 mM ATP, and 10 mM MOPS (pH 7.2) with protease inhibitor cocktail (Roche). The lysate was centrifuged at 48,000 × *g* for 30 min and the clarified supernatant was allowed to incubate with FLAG-resins for 2 h at 4 °C. Bound proteins were washed with buffer containing 10 mM MOPS, 150 mM NaCl, 2 mM ATP, and 0.1 mM EGTA (pH 7.2) for three times. The protein was eluted by adding 300 μg/ml FLAG peptide (GenScript). Eluted proteins were dialyzed overnight against high salt buffer containing 10 mM MOPS, 500 mM NaCl, 0.1 mM EGTA, 2 mM MgCl$_2$, and 1 mM DTT (pH 7.2). Myosins were further concentrated by low speed centrifugation (4000 × *g*, 15 min) with Amicon filter units (Millipore Sigma) and flash frozen with liquid nitrogen for future use. Skeletal muscle actin was purified from rabbit skeletal muscle[66]. 10% Biotinylated F-actin was prepared by mixing unlabled G-actin with biotinylated G-actin (Cytoskeleton Inc, AB07) followed by polymerization in KMEI buffer (50 mM KCl, 2 mM MgCl$_2$ 1 mM EGTA, 10 mM DTT, and 10 mM MOPS (pH 7.4). F-actin was labeled stoichiometrically with fluorescent phalloidin (Thermo Fisher Scientific)[67].

One micromolar of HaloTag-myosin-7a was labeled by incubating with 2 μM total HaloTag ligand overnight in 500 mM NaCl, 20 mM MOPS, 5 mM MgCl$_2$, 0.1 mM EGTA, 10 mM DTT (pH 7.4). Excess dye was removed using Zeba Spin desalting columns (Thermo Fisher Scientific, 7 K molecular weight cutoff, using Buffer Exchange procedure detailed in the manufacturer protocols). For the triple labeling experiment, a single solution of HaloTag-myosin-7a was labeled simultaneously with a mixture of 3 dyes (Promega, TMR, Alexa Fluor 488 & Alexa Fluor 660), each at a concentration of 0.66 μM. A parallel experiment using separately labeled myosins and subsequent mixing yielded the same results.

**Single-molecule motility assays**. Biotinylated polyethylene glycol (biotin-PEG) (Laysan Bio) treated coverslips were created following a previously described protocol[67] with slight modifications. Coverslips were washed and sonicated sequentially (10 min each) in 2% Hellmanex (Sigma) followed by distilled water, and finally 100% ethanol. Coverslips were then thoroughly dried with a stream of filtered air and plasma cleaned for 10 min in argon. The biotin-PEG mixture (2 mg/ml mPEG-silane and 10 μg/ml biotin-PEG-silane combined in 80% ethanol, pH 2.0 using HCl) was added to the surface of coverslips followed by overnight incubation at 70 °C. Coverslips were then rinsed thoroughly with distilled water and used in the creation of flow chambers[68].

10 mg/ml BSA in 150 motility buffer (150 MB) (150 mM KCl, 20 mM MOPS, 5 mM MgCl$_2$, 0.1 mM EGTA, pH 7.4) was added to the flow chamber and incubated for 3 min. Chambers were then washed with 4 chamber volumes of 150 MB. NeutrAvidin (Fisher Scientific, 2 mg/ml in 150 MB) was added to the flow chamber in two successive steps and incubated for 3 min each time. Chambers were then washed with 4 chamber volumes of 150 MB before the addition of 10% biotinylated, phalloidin labeled actin (150–200 nM), which was incubated for 3 min. Chambers were washed with 4 chamber volumes of 150 MB and experiments were then initiated by adding 25 μl of final buffer (150 mM KCl, 20 mM MOPS, 5 mM MgCl$_2$, 0.1 mM EGTA, 2 mM ATP, 50 mM DTT, 100 mg/ml glucose oxidase, 40 μg/ml catalase, 2.5 mg/ml glucose, pH 7.4) containing 20 nM myosin-7a (and 20 nM M7BP when indicated).

Movies were collected on an inverted Nikon Eclipse Ti-E microscope with an H-TIRF module attachment, a CFI60 Apochromat TIRF 100× Oil Immersion Objective Lens (N.A. 1.49, W.D. 0.12 mm, F.O.V 22 mm) and an EMMCD camera (Andor iXon Ultra 888 EMCCD, 1024 × 1024 array, 13 μm pixel). The excitation light source was a Nikon LU-N4 Laser Unit equipped with four lasers (405, 488, 561, and 640 nm).

Where required, movies were drift corrected using the ImageJ/FIJI plugin Image Stabilizer. Processive runs were analyzed using the TrackMate plugin for ImageJ. TrackMate settings were: LoG detector (estimated blob diameter 0.5–0.7 μm, threshold: 40–100), initial threshold: not set, view: HyperStack Displayer, filters on spots: not set, tracker: Simple LAP tracker (Linking max distance: 0.2 μm, gap-closing max distance: 0.2 μm, gap-closing max frame gap: 2), filters on tracks: Duration of track (above 30 s), track displacement (above 0.39 μm), velocity standard deviation (below 0.3). Raw data were obtained from the Analysis tab of the TrackMate plugin and imported into Prism 7 for analysis. Mean straight line speed was used as the output velocity. Gaussian fits were used to determine average velocity. Characteristic run lengths and run durations were determined via exponential fits of the corresponding histograms.

For analysis of fluorophore numbers per moving spot, movies were initially background corrected using an ImageJ macro as follows: for each frame in each channel duplicate the image, determine the mode pixel value, set all pixel values above the mode value to the mode value, gaussian blur the resulting image using Sigma = 100. The resulting smoothed background image was then subtracted from the original image to produce the background subtracted image. All images were recombined into a hyperstack. In order to allow tracking of particles from any individual channel, a new channel was created using the highest pixel value from any channel (not including actin) for each frame. This combined channel was used as the TrackMate input channel. The multi-channel version of TrackMate was then used to determine an intensity for each fluorophore in any moving spot. To calculate an individual fluorophore intensity for each wavelength, kymograph

analysis was used to identify particles which underwent complete photobleaching in the channel of interest (but remained visible in at least one other channel). Using a rounded selection area of the same size as that used in the TrackMate analysis, the spot intensity, immediately prior to the final photobleaching event, was measured. A minimum of 25 such events was measured and averaged to produce an individual fluorophore intensity for each channel. A python script was used with the TrackMate results files to extract the total intensity of each moving spot in each channel in the second frame of the detected run (to avoid partial exposure effects in the first frame). This value was divided by the individual fluorophore intensity to give the number of each molecule per moving complex. Data were imported into GraphPad Prism and plotted as histograms with log-normal fits.

**Actin gliding motility assays**. The sliding actin in vitro motility assay was performed following a standard protocol[69] with minor modifications. Full-length myosin-7a was bound to the coverslip surface in high salt buffers (0.2 mg/ml myosin in 500 mM KCl, 20 mM MOPS, 5 mM MgCl$_2$, 0.1 mM EGTA, 1 mM DTT, pH 7.4). The actin motility was examined in the final assay buffer (150 mM KCl, 20 mM MOPS, 5 mM MgCl$_2$, 0.1 mM EGTA, 1 mM ATP, 50 mM DTT, 2.5 mg/ml glucose, 100 µg/ml glucose oxidase, 40 µg/ml catalase, pH 7.4) in the absence or presence of 500 nM M7BP at room temperature. Movies were collected on an inverted Nikon Eclipse Ti-E microscope with an H-TIRF module attachment, a CFI60 Apochromat TIRF 100× Oil Immersion Objective Lens (N.A. 1.49, W.D. 0.12 mm, F.O.V 22 mm) and an EMMCD camera (Andor iXon Ultra 888 EMCCD, 1024 × 1024 array, 13 µm pixel). 100 ms exposures were acquired every 30 s for 30 min. Four movies were collected and analyzed for each condition. Where required, movies were drift corrected using the ImageJ/FIJI plugin Image Stabilizer.

Motility was quantified using the FAST program[70]. A tolerance filter of 33% was used to exclude intermittently moving filaments and a minimum velocity filter of 0.1 nm/s was used to exclude stuck filaments.

**Actin co-sedimentation assay**. 0.8 µM myosin-7a and M7BP or *Drosophila* myosin-5 were incubated with 4 µM filamentous actin in buffers contain 150 mM NaCl, 10 mM MOPS, 2 mM MgCl$_2$, 0.1 mM EGTA, pH 7.4 for 30 min at room temperature. The samples were then centrifuged at 100,000×g for 20 min at 20 °C (Beckman Optima Ultracentrifuge). Supernatant and pellet fractions were separated with particular caution and examined on a 4–12% Bis-Tris gel (Life Technologies). The gel was stained with SimpleBlue (Invitrogen) and scanned with an Odyssey imaging system (LI-COR).

**Negative stain electron microscopy and image processing**. Proteins were diluted to 40 nM in 150 mM NaCl, 10 mM MOPS, 2 mM MgCl$_2$, 0.1 mM EGTA, 100 µM ATP (pH 7.0). Samples were applied to UV treated, carbon coated EM grids and stained immediately using 1% uranyl acetate. Micrographs were recorded on a JEOL 1200EX microscope using and AMT XR-60 CCD camera at a nominal magnification of ×60,000. Reference-free image alignments and classification were conducted using SPIDER software. For M7BP, 4745 particles were picked from 160 micrographs. Following an initial alignment and classification to remove poorly aligned particles, a second alignment of 2816 particles was classified into 100 classes using K-means classification. For myosin-7a, 4289 particles were selected from 100 micrographs. Following an initial alignment and classification to remove poorly aligned particles, a second alignment of 2800 particles was classified into 100 classes using K-means classification. For myosin-7a + M7BP, 3986 particles were selected from 200 micrographs. Images were aligned and classified into 100 classes using K-means classification. For each dataset, 20 representative classes were selected to produce the montages shown in the main figures.

**Single-molecule mass photometry**. Single-molecule landing assays, data acquisition and image processing were performed as previously described[42,43] using the Refeyn One$^{MP}$ mass photometer. Briefly, microscope coverslips (#1.5 24 × 50 mm and 24 × 14 mm, Fisher Scientific) were cleaned consecutively with isopropanol and water then dried using a stream of clean nitrogen. Pairs of stacked coverslips were used with double sided tape to assemble simple flow chambers. Twenty nanomolar of each myosin-7a or M7BP samples or a mixture of 20 nM M7BP and 10 nM myosin-7a were flowed into the chamber in buffer containing 150 mM NaCl, 2 mM MgCl$_2$, 0.1 mM EGTA, 10 mM MOPS (pH 7.2). Data were collected in a ~3 µm × 10 µm field of view at an acquisition rate of 1 kHz for 100 s. All measurements were carried out at room temperature (~24 °C). Images were processed using manufacturer supplied software (Refeyn, UK). The conversions between molecular mass and interferometric contrast were calibrated with protein standards of known molecular weight. For each sample being measured, a histogram was made and fitted with Gaussians. Fitted centers of these Gaussians and the corresponding masses that they were assigned to were plotted and fitted to a straight line. The fields of view shown in Fig. 1G–I are individual frames taken from movies created using the rolling background subtraction method described in Young et al.[42]

**Bio-layer interferometry**. Binding analyses between myosin-7a and M7BP was performed with Octet RED96 (ForteBio, Pall Corporation). Streptavidin (SA) dip-and-read biosensors (ForteBio) were used for all interaction studies. 20 µg/ml biotinylated GFP antibody (Rockland, 600-406-215) were first coated to the

biosensor surfaces via biotin-streptavidin interactions. Then, 10 µg/ml GFP-tagged myosin-7a was adhered in a "tail-up" orientation by binding to the surface-bound antibodies with its N-terminal GFP moiety. Finally, 0–25 nM M7BP were incubated with immobilized myosin-7a for 10 min followed by 20-min dissociations. All interactions were carried out in buffers containing 150 mM NaCl, 2 mM MgCl$_2$, 0.1 mM EGTA, 10 mM MOPS, 1% BSA (pH 7.2). The experimental temperature is 30 °C.

**Steady-state ATPase assay**. Steady-state ATPase activities were measured in Cary 60 spectrophotometer (Agilent Technologies) at 25 °C in buffers containing 10 mM MOPS, 1 mM ATP, 50 mM NaCl, 2 mM MgCl$_2$, 0.1 mM EGTA (pH 7.2). Data were collected using Cary WinUV software and plotted by Graphpad Prism 7.0. The buffers also contained an NADH-coupled, ATP-regenerating system including 40 units/ml lactate dehydrogenase, 200 units/ml pyruvate kinase, 200 µM NADH and 1 mM phosphoenolpyruvate. 100 nM myosin-7a, 100 nM M7BP, and 20 nM skeletal myosin HMM were included as indicated. The rate of ATP hydrolysis was measured from the decrease in absorbance at 340 nm caused by the oxidation of NADH.

**Cell growth, plasmid DNA transfections, and ER labeling**. *Drosophila* S2 cells were maintained in Schneider's *Drosophila* medium (Thermo Fisher Scientific) supplemented with 10% fetal bovine serum (Sigma) at 27 °C. F-tractin was a gift of Dr. Xufeng Wu and Dr. John A. Hammer (NHLBI). GFP-myosin-7a constructs, M7BP-mCherry and F-tractin-mCherry were inserted into a modified pAc5 vector with standard cloning techniques (Infusion, Clontech) and transfected into cells at 100–250 ng/ml (200–500 ng DNA per unique plasmid for 2-ml culture). Transfection was performed using a non-liposomal transfection reagent (Effectene, QIAGEN). At ~48 h of post-transfection, cells were transferred to chambered cover-glass (Lab-Tak) that had been coated with 0.5 mg/ml concanavalin A (Sigma) for imaging. All images were collected between 1–3 h after cell adhered to Con A substrate. For ER labeling, adherent S2 cells were incubated with 1 µM ER-Tracker Red (Molecular Probes) for 20 min and then subjected to imaging in fresh culture medium.

**Live-cell microscopy and imaging process**. Confocal and Airyscan imaging were performed on a Zeiss LSM 880 Airyscan microscope equipped with a 63 × 1.4 NA objective. For Airyscan, raw data was processed using Airyscan processing in "auto strength" mode (mean strength ± S.D. = 5.5 ± 1.3) with Zen Black software version 2.3. TIRF-SIM imaging was performed on a Delta Vision OMX (GE) equipped with an Olympus 60× /1.49 NA TIRF objective. Raw images were deconvolved using Softworx (Applied Precision). A Wiener Filter Constant of 0.005 was used for every frame of processing. Linear adjustments for contrast and brightness were made to images using ImageJ. Colocalization analysis was performed using an ImageJ Macro Colocalization. A high pass filter of 10 was applied to the image stacks used for the kymograph analysis in Fig. 6C.

**Affinity pull down assay**. Adult fly heads were dissected and homogenized in lysis buffer containing 150 mM NaCl, 0.1% Triton-X, 1 mM EGTA, 20 mM HEPES and protease inhibitor cocktails (Roche), pH 7.5. The tissue extracts were centrifuged at 1000 × g for 10 min. The pull down was then performed by incubating the supernatant with purified FLAG-tagged myosin-7a protein that had been immobilized on FLAG-resins for 1 h with constant rotation. The resins were washed three times and all bound proteins were eluted by adding 1× SDS sample buffer (Invitrogen). All samples were analyzed by SDS/PAGE and M7BP immunoblotting (1:200 for M7BP custom antibody produced by YenZym, 1:10,000 for LI-COR secondary antibody IRDye 800CW, P/N: 926-32211).

**Antibodies and immunofluorescence**. The isolated imaginal discs of third instar larvae were fixed with 4% paraformaldehyde, solubilized in 0.1% Triton-X, PBS solution and then immunostained. The myosin-7a antibodies were raised and conjugated with Alexa-Fluor 488 (Invitrogen, Z25302) and used at a 1:1000 dilution. The M7BP antibodies were custom antibodies produced by SDIX and used at a 1:1000 dilution. Alexa-Fluor conjugated secondary antibodies were purchased from Life Technologies and used at a 1:500 dilution (Cat. A-11036). Samples were mounted in the VECTASHIELD mounting media (Vector Laboratories) and imaged on a Zeiss LSM 780 microscope.

**Reporting summary**. Further information on research design is available in the Nature Research Reporting Summary linked to this article.

## Data availability
All relevant data supporting the finding of this study are available from the authors upon request. Source data are provided with this paper.

## Code availability
The custom macros used for the single-molecule analysis will be made available upon request.

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

## Acknowledgements

We thank the Electron Microscopy, Light Microscopy, Proteomics and Biophysics Core Facilities of the National Heart, Lung, and Blood institute (NHLBI) for their support, advice, and the use of facilities. We thank the laboratory of Philipp Kukura for the mass photometry technology. We thank Fang Zhang for her help with reagent and actin preparation. R.L. is supported by a postdoctoral fellowship from NHLBI (Lenfant Bio-medical Fellowship). This work is also supported by the Intramural Research Program of the NHLBI, NIH (HL006049 to J.R.S.).

## Author contributions

R.L., N.B., Y.Y., and J.R.S. designed the research. R.L., N.B. and Y.Y. performed the experiments and analyzed the experimental data. C.B., A.H., Y.T., and V.S. contributed to the experimental work and data analysis. R.L., N.B., and J.R.S. wrote the manuscript.

## Competing interests

The authors declare no competing interests.
