## [Peer Review File · Nature Communications]

REVIEWER COMMENTS

Reviewer #1 (Remarks to the Author):

Head-tail autoinhibition is a widely used mechanism for regulating the activity of myosin (and kinesin) family members. A key question in the field is how are the motors switched on and, for a number of myosins, induced to dimerize. Myosin-7a (crinkled, ck) is a functionally important motor protein in *Drosophila*. The cellular role of myosin-7a and how its activity is controlled is poorly described. However, the myosin is known to be regulated by autoinhibition and here the authors identify a tail binding protein (M7BP) that both blocks head-tail interaction and induces dimerization of myosin-7a. Ectopic co-expression of the two proteins in cultured fly cells (S2) induces dramatic bundling of actin and the formation of actin-filled protrusions that are somewhat filopodia-like. Expression of a forced dimer of the motor itself induces formation of thin protrusions that resemble filopodia. *In vitro* motility assays show that it is a slow motor and are consistent with the existence of a weak actin binding site in M7BP that is available upon binding to myosin-7a. The authors propose that both myosin-7a and M7BP exist in autoinhibited states that upon a signal or interaction between the two opens them up, then induces dimerization and activation of the motor.

The results potentially represent an important step towards understanding how the activity of a myosin is regulated by a binding partner. The characterization of the motor and its binding partner, and motility of the motor alone or in complex with M7BP is nicely executed. The main question is whether M7BP is truly a physiologically relevant myosin-7a partner since the interaction was identified from a yeast two-hybrid screen. The biochemical data do strongly support the idea of a meaningful interaction, but *in vivo* data is needed. It would be useful to know if M7BP mutants have phenotypes similar to myosin-7a (ck) mutants. While it appears that M7BP (annotated in FlyBase as CG43340) has been identified in a few mutant screens, the phenotypes are described in a largely general manner and it is difficult to know much about what the actual function of the protein based on the available information. The authors present limited immunostaining data showing that both proteins are present in the eye imaginal disc. However, there is little, if any co-localization of myosin-7a and M7BP in the images provided and there is no supporting interaction data such as a co-IP from tissue. Admittedly it can be tricky to see co-localization or association between two proteins that may only both be found in a limited set of tissues or cell types and might even require a specific signaling event to interact. The tight interaction between M7BP and myosin-7a do suggest that these proteins would surely bind to each other *in vivo*. However, absent some supporting data showing interaction between the two in a native context one cannot be sure of the overall significance of the results presented here. The addition of at least stronger co-localization data from tissues where the two proteins are known to be expressed and where myosin-7a is active would help to establish the importance of the interaction of this motor with M7BP.

Detailed comments -

1) The mass photometry measurements show that the myosin-7a and M7BP form a 1:1 complex and not a 2:2 complex. This is at odds with the Halo tag data that support myosin-7a forming a dimer and

moving along actin filaments in the presence of dimeric M7BP. These results suggest to the authors that M7BP exists in an autoinhibited state, similar to myosin-7a, and that accounts for minimal or no dimerization or even actin binding on its own.

Have the authors looked at the dimerization and actin binding of deletion mutants to test this proposal? Also, in the actin binding assay (Fig 1C) it does appear that some of the M7BP does co-sediment with actin (more than background in the M7BP alone lanes). Has the binding been measured directly?

2) The protrusive structures observed in cells co-expressing both full-length myosin-7a and M7BP are referred to as filopodia but their overall appearance is more irregular than a typical filopodium. It would seem that they should rather be referred to as 'filopodia-like'. In fact, they strikingly resemble the actin-based protrusions observed by Masters & Buss (PNAS, 2017) who expressed a 'reversed' Myo6, a non-filopodial myosin, that moved towards the plus ends of actin in mammalian cells in a dimerization-dependent manner. Based on this experiment it is difficult to ascribe filopodia-forming activity for myosin-7a, but it does provide support for the ability of a dimerized motor to bundle and slide actin filaments *in vivo*. In contrast to the seemingly irregular protrusions seen in S2 cells expressing myosin-7a and M7BP, a dimerized motor region of myosin-7a does induce what appear to be more normal-looking (not overly branched or bent) filopodia-like protrusions from the periphery of S2 cells (Fig S5C). For both the dimerized motor alone and the myosin-7a + M7BP expressing cells how efficiently are filopodia made (i.e. fraction of cells and maybe even number of protrusions)? Have the authors stained these protrusions to look for other filopodial proteins (fascin, formin, etc)?

3) The level of both myosin-7a and M7BP expressed in S2 cells seems quite high, although it is difficult to know how much is present simply based on the micrographs presented. Do the authors have any information about how much protein is expressed in the cells that make filopodia-like extensions? Is there a concentration-dependence to the activity (i.e. higher expressing cells make protrusions while lower expressing cells do not).

Minor points -

The authors state (line 66) that "... genetic screens showed that CG43340 is vital for *Drosophila* embryogenesis." but they do not provide a reference in support of this statement.

The pH of the buffers is not always clear based on how they are written out.

Little information is provided about M7BP itself - although it appears to have a Rab binding domain at its N terminus, similar to other myosin binding proteins such as MyRIP (that is a human Myo7A binding partner) and melanophilin (a human Myo5 binding partner). Are there any vertebrate homologs of M7BP?

Reviewer #2 (Remarks to the Author):

Dear Editor,

Please find enclosed the revision of the MS "NCOMMS-20-15325-T" entitled "Molecular Mechanism of Myosin-7a Translocation and Actin Bundle Assembly: Insights from A New Binding Protein" by Liu et al.

The authors characterized a new Myosin-7a binding protein M7BP. M7BP binds to Myosin-7a activating its ATPase activity and allowing the formation of a complex necessary for efficient motility along actin filaments. In addition, the authors reported that Myosin-7a-M7BP complex is able to induce actin bundle formation both in vitro and in cell.

This is a very interesting and complete study and the manuscript can be published as it is.

I have however some questions that the authors may want to address in the revised version of their manuscript.

1) The authors mentioned that the stoichiometry of the myosin-7A-M7BP is 1:1 but did not discuss why they did not observe higher order oligomers. Same for the 2:2 processive complexes.

2) How much the bundling activity depends on the myosin7a ATPase activity?

3) Since actin bundling by the Myosin-7-a-M7BP complex is followed by actin filament sliding maybe the authors should discuss the overall stability of these actin bundles.

4) According to the EM data in Figure 1 can the authors speculate about the distance between actin filaments inside the bundles and how other crosslinkers (fascin) may bind or not to these bundles.

5) I was less convinced by the Figure 5 than the Figure 6. Maybe the authors should consider moving Figure 5 in supplemental Figure and keep only Figure 6 at the main figure.

Reviewer #3 (Remarks to the Author):

This manuscript reports the discovery of a *Drosophila* myosin-7a binding protein M7BP, and an in-depth study using a range of modern biophysics tools on the molecular mechanism of how M7BP assembles myosin-7a into a motile complex that enables cargo translocation and actin cytoskeletal remodeling. M7BP was discovered using yeast-two-hybrid assay. Purified myosin-7a and M7BP were produced using baculovirus/Sf9 system. The binding affinity of M7BP to myosin-7a was measured by BLI. The shape and size of bound and unbound proteins were studied by electron microscopy and single-molecule mass photometry. Single-molecule mass photometry is a recently developed optical method based on interferometric scattering microscopy (iSCAT), which is able to quantify single protein mass based on the image intensity without the need of labeling. In addition, single molecule

motility imaged by TIRF shows that M7BP promote myosin07a binds and moves along actin filaments at ~ 7.7 nm/s. Quantitative single-molecule fluorescence analysis reveals the most common structure of the moving complex are 2:2 myosin-7a-M7BP dimers. Furthermore, In vitro reconstitution of actin filaments with present of myosin-7a-M7BP complexes were found to form active filament alignment. Finally, cells transfected with both myosin-7a and M7BP show extensive filamentous network and long-lived filopodial protrusions. Point mutation control experiment shows that M7BP subunit is essential for actin network reorganization. Super-resolution TIRF visualized myosin-7a-M7BP complex assembles actin bundles and transports along actin-rich protrusions in cells. The studies are well designed with sufficient control experiments to confirm/validate the results. The finding is interesting and impactful for understanding the mechanisms of the formation of actin protrusions. The manuscript is well written. A minor revision to add some missing experimental details will help the reader to understand and reproduce the work:

Some experimental details for single protein mass photometry is missing. Since it is a relatively new method, a detailed description on how the experiments were carried is essential to help the reader understand the results and reproduce the work. For examples: what was the power density of the light, frame rate of recording, exposure time, field of view for single protein mass photometry results shown in Fig 1G-I and Fig S3C? (in ref 40, different sized proteins use different settings) What was the flow rate and temperature? It will also helpful to show representative iSCAT images and intensity line profiles of the myosin-7a, M7BP proteins and the complex.

Reviewer #1

Head-tail autoinhibition is a widely used mechanism for regulating the activity of myosin (and kinesin) family members. A key question in the field is how the motors switched on and, for a number of myosins, induced to dimerize. Myosin-7a (*crinkled, ck*) is a functionally important motor protein in *Drosophila*. The cellular role of myosin-7a and how its activity is controlled is poorly described. However, the myosin is known to be regulated by autoinhibition and here the authors identify a tail binding protein (M7BP) that both blocks head-tail interaction and induces dimerization of myosin-7a. Ectopic co-expression of the two proteins in cultured fly cells (S2) induces dramatic bundling of actin and the formation of actin-filled protrusions that are somewhat filopodia-like. Expression of a forced dimer of the motor itself induces formation of thin protrusions that resemble filopodia. In vitro motility assays show that it is a slow motor and are consistent with the existence of a weak actin binding site in M7BP that is available upon binding to myosin-7a. The authors propose that both myosin-7a and M7BP exist in autoinhibited states that upon a signal or interaction between the two opens them up, then induces dimerization and activation of the motor. The results potentially represent an important step towards understanding how the activity of a myosin is regulated by a binding partner. The characterization of the motor and its binding partner, and motility of the motor alone or in complex with M7BP is nicely executed.

Thank you for the positive comments.

The main question is whether M7BP is truly a physiologically relevant myosin-7a partner since the interaction was identified from a yeast two-hybrid screen. The biochemical data do strongly support the idea of a meaningful interaction, but *in vivo* data is needed. It would be useful to know if M7BP mutants have phenotypes similar to myosin-7a (*ck*) mutants. While it appears that M7BP (annotated in FlyBase as CG43340) has been identified in a few mutant screens, the phenotypes are described in a largely general manner and it is difficult to know much about what the actual function of the protein based on the available information. The authors present limited immunostaining data showing that both proteins are present in the eye imaginal disc. However, there is little, if any co-localization of myosin-7a and M7BP in the images provided and there is no supporting interaction data such as a co-IP from tissue. Admittedly it can be tricky to see co-localization or association between two proteins that may only both be found in a limited set of tissues or cell types and might even require a specific signaling event to interact. The tight interaction between M7BP and myosin-7a do suggest that these proteins would surely bind to each other *in vivo*. However, absent some supporting data showing interaction between the two in a native context one cannot be sure of the overall significance of the results presented here. The addition of at least stronger co-localization data from tissues where the two proteins are known to be expressed and where myosin-7a is active would help to establish the importance of the interaction of this motor with M7BP.

We thank the reviewer for this suggestion. As advised, we have performed an affinity pull-down assay by using the purified myosin-7a protein (immobilized on FLAG-resins) to capture M7BP from the adult fly tissues. We show that M7BP was present in fly heads and can bind to and be eluted along with myosin-7a (new Fig.S1E). This result is consistent with the previous mass spectrometry studies (Aradska J. et al. *Proteomics*, 2015) showing that CG43340, isoform C is a protein enriched in fly heads. Together, these results argue strongly that M7BP exists in *Drosophila* tissues and is a *bona fide* binding partner for myosin-7a.

CG43340 was shown to be preferentially expressed in SOPs (sensory organ precursor cells) in response to proneural signaling (Reeves N & Posakony JW, *Dev Cell*, 2005). Later, Aerts et al. identified that CG43340 is one of the target genes for Atonal, a proneural factor which specifies the switch from undifferentiated pluripotent cells to founding photoreceptor neurons during larval

development (Aerts S et al. *PLoS Biol*, 2010). Myosin-7a is known to function in sensory bristles and the auditory Johnston's organ in *Drosophila*. Similarly, mammalian myosin-7a is involved in the sensory perceptions of hearing and vision. Unpublished data in our lab have also shown that *Drosophila* myosin-7a is involved in the vesicular transport in fly eyes. Taken together, these data suggest that myosin-7a and M7BP likely co-exist and interact in sensory organs where they are both found essential for the development and functions of these sensory neurons.

CG43340 was referred as CG30492 in early publications. This may lead to some overlooking of the existing data about CG43340. We have added this information in the manuscript text to make it clear.

Detailed comments

1) The mass photometry measurements show that the myosin-7a and M7BP form a 1:1 complex and not a 2:2 complex. This is at odds with the Halo tag data that support myosin-7a forming a dimer and moving along actin filaments in the presence of dimeric M7BP. These results suggest to the authors that M7BP exists in an autoinhibited state, similar to myosin-7a, and that accounts for minimal or no dimerization or even actin binding on its own. Have the authors looked at the dimerization and actin binding of deletion mutants to test this proposal?

We thank the reviewer for raising this point. Indeed, we designed multiple deletion constructs of M7BP to test this possibility. Unfortunately, the expressed constructs were unstable *in vitro*, forming large aggregates, and the same was true for constructs expressed in cells. For these reasons, we were unable to parse the different behaviors of M7BP by deletions of individual domains. With high resolution structures for M7BP we will likely be able to design better deletion constructs, but at the current time we do not have these structures.

Also, in the actin binding assay (Fig 1C) it does appear that some of the M7BP does co-sediment with actin (more than background in the M7BP alone lanes). Has the binding been measured directly?

We have done experiments to measure the affinity of M7BP for actin using co-sedimentation assays. As shown here, while centrifuging with actin caused some M7BP pelleting, this effect essentially plateaus and further increases in actin concentration did not result in sufficient binding for a K_d to be calculated.

The results could be interpreted as a minority of M7BP which binds with an affinity around $0.5 \mu\text{M}$ whilst the majority has an affinity too weak to measure. This may be related to autoinhibition of M7BP itself but we were unable to get a sufficiently clear answer to warrant inclusion in the manuscript. Attempts to investigate this using a combination of M7BP and myosin tail fragments (to relieve autoinhibition of M7BP) were not possible due to the refractory nature of those fragments.

2) The protrusive structures observed in cells co-expressing both full-length myosin-7a and M7BP are referred to as filopodia but their overall appearance is more irregular than a typical filopodium. It would seem that they should rather be referred to as 'filopodia-like'. In fact, they strikingly resemble the actin-based protrusions observed by Masters & Buss (PNAS, 2017) who expressed a 'reversed' Myo6, a non-filopodial myosin, that moved towards the plus ends of actin in mammalian cells in a dimerization-

dependent manner. Based on this experiment it is difficult to ascribe filopodia-forming activity for myosin-7a, but it does provide support for the ability of a dimerized motor to bundle and slide actin filaments *in vivo*. In contrast to the seemingly irregular protrusions seen in S2 cells expressing myosin-7a and M7BP, a dimerized motor region of myosin-7a does induce what appear to be more normal-looking (not overly branched or bent) filopodia-like protrusions from the periphery of S2 cells (Fig S5C). For both the dimerized motor alone and the myosin-7a + M7BP expressing cells how efficiently are filopodia made (*i.e.* fraction of cells and maybe even number of protrusions)?

The complex's ability to induce filopodia in S2 cells was modest compared to that of myosin-7a zippered dimer (as described in the original manuscript and in the quantification shown in the new Fig.S5F). We attributed this difference to the complex's actin-bundling activity and its slower plus-end motility than the motor on its own as we have directly shown in Fig. 4A-C and Fig.S4. While it remains unclear how the complex's actin-bundling activity facilitates the functions of myosin-7a in specific actin-bundle structures in tissues, we speculate that filopodia initiation may not be the primary function of this complex regarding its slow motility and actin-crosslinking ability in contrast to the dynamic nature of filopodia. Instead, the mechanical properties of this complex appear to be optimized for myosin-7a operating in particular actin bundles which are much more stable and rigid than filopodia such as those in bristles and scolopidia (the mechanosensory receptor in Johnston's organ).

We agree with the reviewer that it is more accurate to refer those protrusions as "filopodia-like" extensions. We have incorporated the new quantification figure and updated the text accordingly.

Have the authors stained these protrusions to look for other filopodial proteins (fascin, formin, *etc.*)?

Immunostaining in S2 cells is mainly limited by the availability of *Drosophila* antibodies. While we could not find any commercial antibodies that can react with *Drosophila* formins, we have attempted to stain the protrusion structures with the fascin and Ena/VASP antibodies: sn7C and 5G2 (previously used in the publication from Rogers S. *et al.* reference 51). As shown here, *enabled* displayed a continuous distribution along the periphery in non-transfected S2 cells, whereas in cells expressing the myosin-7a-M7BP complexes, *enabled* appeared to relocate and cluster in places where the protrusions were observed to form (indicated by yellow arrows).

The fascin antibody sn7C did not produce any specific staining in the cultured S2 cells. However, since we currently have little information about this antibody or the localization of fascin in S2 cells, we are reluctant to conclude whether fascin is present in those protrusion structures or not. A more sophisticated assay such as knock-in a GFP-tag to endogenous fascin and either directly observing or immunostaining with an established GFP antibody can be used to address this question.

3) The level of both myosin-7a and M7BP expressed in S2 cells seems quite high, although it is difficult to know how much is present simply based on the micrographs presented. Do the authors have any information about how much protein is expressed in the cells that make filopodia-like extensions? Is there a concentration-dependence to the activity (*i.e.* higher expressing cells make protrusions while lower expressing cells do not).

All the transgenes delivered to S2 cells were driven by the same promoter Ac5 for a constitutive expression. It is clear that the actin bundles and filopodia-like extensions were only induced when cells coexpressing both full-length myosin-7a and M7BP. By contrast, expressing either of them individually does not produce this effect, even when two times more DNA were transfected. We therefore conclude that the cytoskeletal remodeling is a specific cellular event driven by the myosin-7a-M7BP complex, in agreement with our findings *in vitro*.

Regarding the myosin-7a and M7BP co-expression itself, we have used low (200 ng of DNA per unique plasmid) and high (500 ng of DNA per unique plasmid) transfection levels throughout our experiments. Overall the two conditions are very similar and we were able to observe the extension of filopodia-like protrusions in both cases. It is possible that the actin-crosslinking activity within the cell body may counteract the extension of filopodia-like protrusions in the complex-transfected S2 cells. To investigate which activity would be favored by further manipulating the expression level in cells is an interesting direction and may be tested in a dedicated study focusing on this particular question.

We have updated this information in the sections of experimental procedures.

Minor points

The authors state (line 66) that "... genetic screens showed that CG43340 is vital for *Drosophila* embryogenesis." but they do not provide a reference in support of this statement.

We have added references in the text.

The pH of the buffers is not always clear based on how they are written out.

We have adopted a consistent format of buffer writing where pH is placed at the end of the buffer contents to indicate the pH of the entire buffer.

Little information is provided about M7BP itself - although it appears to have a Rab binding domain at its N-terminus, similar to other myosin binding proteins such as MyRIP (that is a human Myo7A binding partner) and melanophilin (a human Myo5 binding partner). Are there any vertebrate homologs of M7BP?

A homology search of the human proteome reveals that M7BP is most closely related to the human proteins MyRIP, melanophilin, rabphilin *etc.* Although the primary sequence conservation (particularly the C-terminal area) falls below that which would classify M7BP as a *Drosophila* homolog to MyRIP, it is clear that M7BP belongs to a class of proteins which share common features such as Rab binding domains and are often associated with regulation of myosin motility. We have updated the text to make this clear.

Reviewer #2:

Dear Editor, please find enclosed the revision of the MS "NCOMMS-20-15325-T" entitled "Molecular Mechanism of Myosin-7a Translocation and Actin Bundle Assembly: Insights from A New Binding Protein" by Liu et al. The authors characterized a new Myosin-7a binding protein M7BP. M7BP binds to Myosin-7a activating its ATPase activity and allowing the formation of a complex necessary for efficient motility along actin filaments. In addition, the authors reported that Myosin-7a-M7BP complex is able to induce actin bundle formation both *in vitro* and in cell. This is a very interesting and complete study and

the manuscript can be published as it is. I have however some questions that the authors may want to address in the revised version of their manuscript.

1) The authors mentioned that the stoichiometry of the myosin-7A-M7BP is 1:1 but did not discuss why they did not observe higher order oligomers. Same for the 2:2 processive complexes.

The absence/rarity of oligomers (greater than 1:1) in mass photometry data and EM suggests that the process of forming these is dependent on the presence of actin. The exact mechanism is elusive and as stated in the discussion may involve a tripartite interaction between myosin-7a, M7BP and actin. It is also possible that the relatively long lifetime of myosin-7a on actin, even in weak binding states, allows for proximity induced enhancement of the interaction between molecules. We have updated the discussion to state clearly that higher order oligomerization requires the presence of actin.

In the case of motile complexes, it can be seen that higher order oligomers do indeed exist, and the multiple color HaloTag myosin assay was designed to test this possibility directly. Individual moving complexes can be observed which contain all three myosin colors. The calculated oligomer distributions from TIRF motility assays show that whilst higher order complexes do exist, the 2:2 oligomer is the most common and higher order complexes are progressively less likely (with an exponential decay in the probability of observing N molecules in a complex). The result suggests the existence of something more complicated than a simple dimerization interface and high resolution structures will be needed to better determine the nature of this interface. We have included these points in the discussion.

2) How much the bundling activity depends on the myosin7a ATPase activity?

We have conducted electron microscopy experiments which show that M7+M7BP will bundle actin in the absence of ATP (see examples below). In this limited sense, ATPase is not a strict requirement for bundling. That being said, any myosin or myosin complex which has more than one motor is expected to bundle actin in the absence of ATP, due to the high affinity of the ATP-free motor for actin.

We have not attempted to create a loss of function mutant for the myosin in cells for the following reasons: A strong binding mutant would be expected to result in some bundling due simply to an ability to bridge multiple actins. A weak binding mutant would be difficult to interpret regardless of outcome. If no bundling were observed, it would be attributable to the lack of actin binding and if bundling were observed it could be attributed to the relatively high affinity of myosin 7 in the “weakly bound” states. In other words, we expect that the large change to the actin affinity which would result from a loss of function mutant would be the primary causes of any observed effects, rather than a loss of ATPase activity *per se*. We don't currently have a way to uncouple these two properties of the myosin.

What we consider to be most important in this context is that the myosin-7 complex can perform the role of a force generating crosslinker/bundling protein, actively driving the alignment of actin filaments. This specific behavior does require ATPase activity. We have included a more thorough elaboration of this important point in the discussion.

3) Since actin bundling by the Myosin-7a-M7BP complex is followed by actin filament sliding maybe the authors should discuss the overall stability of these actin bundles.

We consider it likely that this type of bundling would be somewhat distinct from bundling caused by the true actin crosslinkers of the cell, such as fascin, filamin, α -actinin *etc*. In the case of motor based bundling the process would be dynamic and the final bundles less well ordered. Following an early stage of motor-based coalescence, the proximity of actin filaments would allow for *bona fide* actin crosslinkers to form more ordered and stable actin bundles. We have not directly tested this hypothesis

as it is beyond the bounds of the current study but this would make for an interesting future study looking at the rates and timing of events and the contribution of different crosslinkers to bundle stability.

4) According to the EM data in Figure 1 can the authors speculate about the distance between actin filaments inside the bundles and how other crosslinkers (fascin) may bind or not to these bundles.

We have previously conducted EM experiments looking at actin bundles formed *in vitro* by M7+M7BP. This was done primarily in an attempt to determine the structure of the crosslinks, which proved unsuccessful due to the heterogenous appearance and variable filament spacing. The bundles in those cases are quite heterogenous in terms of filament-filament distances and lack the periodicity/paracrystalline order described for bundles formed *in vitro* by *bona fide* actin crosslinkers such as fascin, filamin and α -actinin. The apparent filament-filament spacing of actin filaments at the crosslinks is defined by the size and orientation of the complex and is typically in the order of 10-30 nm. It is therefore well suited to bringing actin filaments sufficiently close to enhance bundling by other crosslinkers. Due to the need for brevity in the manuscript and the heterogeneous nature of the bundles formed by myosin, we elected not to include that data in this manuscript. If conditions can be found to grow more homogenous bundles, it may be possible in future to better determine the structure of the crosslinks.

Examples of the loose bundling (1 minute after adding M7+M7BP to actin in the absence of ATP) are shown below.

The ability for myosin to enhance crosslinking *in vitro* would be an interesting addition to the possible study of bundle formation in cells. It seems probable that the effects of myosin bundling and crosslinker bundling would be multiplicative and this could be tested *in vitro* in a dedicated study. In relation to this point, as well as points 2 and 3, we have modified the discussion to make it clear that we envision this to be a process whereby actin filaments are brought together by the active motor complex, resulting in a loosely defined bundle which can then be further stabilized by other static crosslinkers.

5) I was less convinced by the Figure 5 than the Figure 6. Maybe the authors should consider moving Figure 5 in supplemental Figure and keep only Figure 6 at the main figure.

Figure 5 makes a number of points which we think are central to the overall findings and are not covered in Figure 6. It shows that myosin-7a and M7BP are colocalized in the cell in bundles and protrusions and that both myosin-7a and M7BP are required for the formation of bundles. It demonstrates that the bundling behavior requires the presence of the myosin tail and finally that the velocities of the moving complex *in vitro* are similar to those observed in the cell. We feel that moving this information to supplementary would make it more difficult for readers to follow the logic of the manuscript and therefore request that the figure stay intact.

Reviewer #3:

This manuscript reports the discovery of a *Drosophila* myosin-7a binding protein M7BP, and an in-depth study using a range of modern biophysics tools on the molecular mechanism of how M7BP assembles myosin-7a into a motile complex that enables cargo translocation and actin cytoskeletal remodeling. M7BP was discovered using yeast-two-hybrid assay. Purified myosin-7a and M7BP were produced using baculovirus/*Sf9* system. The binding affinity of M7BP to myosin-7a was measured by BLI. The shape and size of bound and unbound proteins were studied by electron microscopy and single-molecule mass photometry. Single-molecule mass photometry is a recently developed optical method based on interferometric scattering microscopy (iSCAT), which is able to quantify single protein mass based on the image intensity without the need of labeling. In addition, single molecule motility imaged by TIRF shows that M7BP promote myosin-7a binds and moves along actin filaments at ~ 7.7 nm/s. Quantitative single-molecule fluorescence analysis reveals the most common structure of the moving complex are 2:2 myosin-7a-M7BP dimers. Furthermore, *In vitro* reconstitution of actin filaments with present of myosin-7a-M7BP complexes were found to form active filament alignment. Finally, cells transfected with both myosin-7a and M7BP show extensive filamentous network and long-lived filopodial protrusions. Point mutation control experiment shows that M7BP subunit is essential for actin network reorganization. Super-resolution TIRF visualized myosin-7a-M7BP complex assembles actin bundles and transports along actin-rich protrusions in cells. The studies are well designed with sufficient control experiments to confirm/validate the results. The finding is interesting and impactful for understanding the mechanisms of the formation of actin protrusions. The manuscript is well written. A minor revision to add some missing experimental details will help the reader to understand and reproduce the work:

Some experimental details for single protein mass photometry is missing. Since it is a relatively new method, a detailed description on how the experiments were carried is essential to help the reader understand the results and reproduce the work. For examples: what was the power density of the light, frame rate of recording, exposure time, field of view for single protein mass photometry results shown in Fig 1G-I and Fig S3C? (in ref 40, different sized proteins use different settings) What was the flow rate and temperature? It will also helpful to show representative iSCAT images and intensity line profiles of the myosin-7a, M7BP proteins and the complex.

We have added more detail describing the mass photometry to the experimental procedures. The flow chamber is used only to introduce samples prior to imaging and samples are not under flow during data acquisition. We have therefore not included a flow rate in the methods but have made the description of the flow chamber clearer. Examples images of each condition have now been added below the histograms in figure 1. The intensity scaling is equivalent for all three of the added images, so that the three can be directly compared, although it should be noted that during actual data collection and analysis (which is performed within the manufacturer's custom software), the intensities of many thousands of such particles are analyzed as proteins land on the imaging surface, in order to produce the histograms shown in the manuscript. The process relies on a rolling background subtraction and a peak detection which identifies the maximum intensity difference relative to the region before landing. As such, there is a temporal element which is missing in single images (and individual line profiles of particles). This is extracted from the raw movies using the multi-step process performed by the manufacturer's software (as described in Young et al, Reference 40) and is required to obtain the precise intensity measurements that ultimately get converted into a molecular mass.

REVIEWERS' COMMENTS<

Reviewer #1 (Remarks to the Author):

The authors have made a good effort to obtain additional data to address the points raised in the original review but in the end were unable to do so because of significant technical difficulties on several fronts. While this is disappointing, the authors have otherwise satisfactorily addressed this reviewer's comments and the lack of the requested data does not in any way detract from the fundamentally interesting contribution presented in this manuscript.

Minor comments -

The first sentence of the Abstract states that Myosin-7a 'promotes actin-bundle protrusions', including filopodia, microvilli and stereocilia.'

Strictly speaking, most of the myosin 7 family members (that should perhaps just be referred to as Myosin-7 instead of Myosin-7a?) play roles in organizing actin-based protrusions such as microvilli and stereocilia. The authors might consider revising this sentence so as not to give the unintended impression that these myosins have roles in the formation of microvilli and stereocilia. Perhaps merging the first two sentences would make their meaning more clear.

Fig S5A - the last panel in the set should be labeled MERGE

Reviewer #2 (Remarks to the Author):

The authors have satisfactorily addressed my comments.

Reviewer #3 (Remarks to the Author):

The revision has addressed all my questions in the first round review, and I believe the manuscript is ready for publication.

Reviewer #1 (Remarks to the Author):

The authors have made a good effort to obtain additional data to address the points raised in the original review but in the end were unable to do so because of significant technical difficulties on several fronts. While this is disappointing, the authors have otherwise satisfactorily addressed this reviewer's comments and the lack of the requested data does not in any way detract from the fundamentally interesting contribution presented in this manuscript.

We thank the reviewer for these comments and the time reviewing the manuscript.

Minor comments -

The first sentence of the Abstract states that Myosin-7a 'promotes actin-bundle protrusions', including filopodia, microvilli and stereocilia.'

Strictly speaking, most of the myosin 7 family members (that should perhaps just be referred to as Myosin-7 instead of Myosin-7a?) play roles in organizing actin-based protrusions such as microvilli and stereocilia. The authors might consider revising this sentence so as not to give the unintended impression that these myosins have roles in the formation of microvilli and stereocilia. Perhaps merging the first two sentences would make their meaning more clear.

As suggested, we have edited and merged the first two sentences in the Abstract to make this more clear.

Fig S5A - the last panel in the set should be labeled MERGE

The last panel in Fig S5A has been labeled as Merge.

Reviewer #2 (Remarks to the Author):

The authors have satisfactorily addressed my comments.

We thank the reviewer for these comments and the time reviewing the manuscript.

Reviewer #3 (Remarks to the Author):

The revision has addressed all my questions in the first round review, and I believe the manuscript is ready for publication.

We thank the reviewer for these comments and the time reviewing the manuscript.